# Microwave-Assisted Degradation of Biomass with the Use of Acid Catalysis

**Grzegorz Kłosowski**, **Dawid Mikulski \*** and **Natalia Lewandowska**

Department of Biotechnology, Kazimierz Wielki University, ul. K. J. Poniatowskiego 12, 85–671 Bydgoszcz, Poland; klosowski@ukw.edu.pl (G.K.); natalialewandowska012@wp.pl (N.L.)

**\*** Correspondence: dawidmikulski@ukw.edu.pl; Tel.: +48-(052)-32–59-218; Fax: +48-(052)-37–67-930

**Abstract:** The aim of the study was to assess the effectiveness of microwave pretreatment combined with acid catalysis in the decomposition of various types of biomass (pine and beech chips and hemp stems). It was clearly demonstrated that sulfuric acid was a catalyst enabling the most effective decomposition of the tested plant biomass, guaranteeing the highest concentrations of simple sugars released. Acid catalysis with 1% *v/v* sulfuric acid combined with microwave radiation provided high glucose concentrations of 89.8 ± 3.4, 170.4 ± 2.4 and 164.6 ± 4.6 mg/g for pine chips, beech chips and hemp stems, respectively. In turn, the use of nitric acid promoted the degradation of hemicellulose, which resulted in high concentrations of galactose and xylose, i.e., 147.6 ± 0.6, 163.6 ± 0.4 and 134.9 ± 0.8 mg/g of pine chips, beech chips and hemp stems, respectively, while glucose levels remained relatively low. It was also demonstrated that the undesirable dehydration of sugars such as glucose and xylose is more pronounced in sulfuric acid than nitric acid processes. The use of $H_2SO_4$ and increased pressure generated 5-hydroxymethylfurfural (5-HMF) and furfural at a concentration of ca. 12 and 6 mg/g, 10 and 45 mg/g and 14 and 30 mg/g, of pine chips, beech chips and hemp shoots, respectively. Our studies confirmed the usefulness of the combined use of microwaves and acid catalysis in the degradation of softwood, hardwood and non-wood plant biomass. It should be emphasized that obtaining high concentrations of released simple sugars (as potential substrates in biosynthesis), while maintaining low levels of toxic by-products (inhibitors), requires precise selection of process parameters such as pressure, exposition time and type of acid catalyst.

**Keywords:** acid catalysis; microwave-assisted degradation; biomass

## 1. Introduction

Worldwide demand for energy increases with the development of the global economy. Currently, energy is mainly produced from fossil fuels, whose resources are limited. Their use generates huge amounts of carbon dioxide, nitrogen and sulfur oxides, and dust that contribute to air pollution. An alternative which would limit these negative effects is the production of energy from renewable sources such as plant-derived raw materials containing carbohydrates [1]. The first generation biofuel production technologies, such as the production of ethanol from raw materials containing starch or sucrose, are well known and widely used on an industrial scale. In the long run, however, the growing use of these raw materials for biofuel production may lead to higher food and feed prices and, consequently, also higher animal product prices. One of the solutions to this problem is the widespread production of second-generation biofuels by conversion of non-food plant biomass [2].

From an economic point of view, the source of biomass used in the conversion process should be widely available and cheap. These requirements are met by waste plant biomass, which includes wood industry waste (shavings, wood chips), agri-food waste (brewer's spent grain, stillage, beet pulp, press cake) and paper industry waste [1,3,4]. The basic parameter determining the usefulness of plant

biomass in the production of biofuels is its composition, in particular the relative content of cellulose, hemicellulose and lignin. The content of lignocellulose components in biomass depends on its type. Softwood biomass contains 45–50% *w/w* cellulose, 25–35% *w/w* hemicellulose and 25–35% *w/w* lignin. Hardwood biomass is made of 40–55% *w/w* cellulose, 24–40% *w/w* hemicellulose and 18–25% *w/w* lignin. Non-wood biomass (mainly grass) contains 20–40% *w/w* cellulose, 35–50% *w/w* hemicellulose and 10–30% *w/w* lignin [5,6]. It should be emphasized, however, that making full use of lignocellulosic biomass requires the application of an effective method of disintegration of its complex structure that would ensure high concentrations of carbohydrates susceptible to the bioconversion process. Many physical, chemical and biological methods for the pretreatment of lignocellulosic biomass have been developed. However, the most effective include combined physicochemical methods using acid catalysis in addition to elevated temperature and pressure [7]. Microwave radiation, i.e., electromagnetic radiation with a frequency of 0.3 to 300 GHz has recently been increasingly used for plant biomass pretreatment. The heating of the reaction medium using microwaves is more energy efficient and faster than conventional heating and provides better control of reaction temperature because the process can be stopped immediately. In addition, it was confirmed that dipole motions caused by microwaves intensified the disruption of hydrogen bonds in cellulose fibers [8]. Microwave radiation is currently successfully used in the process of thermochemical conversion of cellulose, direct conversion of wood waste to levulinic acid, extraction of lignin from plant biomass and production of second generation ethanol [9–12].

The type and concentration of the acid catalyst, apart from the biomass heating method, is a key factor guaranteeing a high level of degradation of cellular structures. In the decomposition of plant biomass, sulfuric, nitric and hydrochloric acid are most often used. The use of dilute sulfuric acid at elevated temperatures resulted in high concentrations of glucose and xylose, i.e., products of cellulose and hemicellulose degradation. However, the effectiveness of acid catalysis strongly depended on the temperature, pressure and duration of the pretreatment procedure [13,14]. Pretreatment of lignocellulose with mineral acids at elevated pressure not optimized for process parameters contributes to increased dehydration of simple sugars and formation of large amounts of toxic by-products such as 5-hydroxymethylfurfural (5-HMF) from glucose and furfural from xylose. Determining optimal biomass degradation parameters using acid catalysts is extremely important to produce hydrolysates with a high content of simple sugars and low concentration of toxic by-products. Such hydrolysates can then be a substrate in biosynthesis or bioconversion processes, e.g., in the production of second generation bioethanol [15,16].

The aim of the present study was to examine the efficiency of microwave radiation in the decomposition of the carbohydrate fraction of softwood, hardwood and non-wood biomass using acid catalysis with 1% *v/v* $H_2SO_4$ or 1% *v/v* $HNO_3$. The use of microwaves for plant biomass decomposition should be preceded by a detailed analysis of the impact of microwave processing parameters (pressure and exposure time) and the appropriate selection of an acid catalyst. Until now, studies have focused mainly on the use of microwaves to degrade non-wood biomass such as wheat and rice straw, which are waste generated by the food industry [17–20]. In the present work, the authors determined the impact of various microwave processing conditions (combinations of pressure, time and type of acid catalyst) on the degradation of polysaccharides contained in waste biomass of raw materials that have not yet been analyzed, such as pine and beech chips (soft- and hardwood) and hemp stems (a non-wood material). The assessment of microwave treatment in the degradation of waste biomass of raw materials resistant to catalytic decomposition can contribute to an effective utilization of agricultural and wood waste for the production of renewable energy using microbiological conversion processes. Previous studies on the impact of microwave pretreatment parameters including pressure, exposure time and type of acid catalyst on the degradation of softwood, hardwood and non-wood biomass, have been very fragmentary. The impact of sulfuric and nitric acid combined with microwave pretreatment on the degradation of cellulose, hemicellulose and the formation of toxic by-products has not yet been described. From a technological point of view, it seems important to define process

parameters that guarantee a high concentration of fermentable sugars that can be used in biosynthesis and bioconversion processes. Previous studies on the combined use of microwave radiation and various acid and alkaline catalysts, or ionic liquids in the pretreatment of plant biomass were carried out on raw materials such as: distillery stillage, sugar cane shoots, rice straw or algae biomass. The results presented in this paper are a step towards the effective microwave-assisted processing of further potential lignocellulosic raw materials, such as softwood, hardwood and non-wood biomass. It should be noted that high process efficiency requires careful optimization of process parameters, separately for each type of raw material. These and previous results achieved for various lignocellulosic materials indicate that it is practically impossible to propose universal conditions for the pretreatment process that could be successfully used regardless of the type of plant biomass. The material-dependent biomass preprocessing is not just the choice of pretreatment conditions that ensure high hydrolysis efficiency. This approach also reduces or eliminates the adverse formation of excessive amounts of inhibitors.

## 2. Results and Discussion

### 2.1. Impact of Microwave Pretreatment Conditions on the Amount of Carbohydrates Obtained from Analyzed Raw Materials

The conducted studies clearly indicated a higher efficiency of sulfuric acid in microwave-assisted decomposition of lignocellulosic biomass (Figure 1). In comparison with nitric acid, sulfuric acid used as a catalyst provided higher glucose, galactose and xylose concentrations. The maximum concentration of carbohydrates was achieved at different pressure values during microwave treatment depending on the acid used. When preprocessing pine and beech chips, the highest glucose concentration ($89.8 \pm 3.4$ mg/g and $170.4 \pm 2.4$ mg/g of biomass, respectively) was obtained with 1% *v/v* $H_2SO_4$, at a pressure of 93 PSI and 15 min exposure time (Figure 1A,C). Microwave-assisted decomposition of cellulose from pine and beech chips was most effective when using sulfuric acid, which was reflected in the highest glucose concentration at 93 PSI. The use of 1% *v/v* $H_2SO_4$ and lower pressure (54 PSI) promoted the decomposition of hemicellulose, which resulted in an increased concentration of galactose and xylose, while a higher pressure of 152 PSI increased sugar dehydration, which resulted in reduced carbohydrate concentration (Figure 1A,C). The highest glucose concentrations from hemp stems (above 144 mg/g of biomass) were obtained using sulfuric acid at 93 and 152 PSI regardless of the exposure time (Figure 1E). In contrast to cellulose, the highest degree of hemicellulose degradation was achieved using 1% *v/v* sulfuric acid, at the lowest pressure of 54 PSI regardless of the biomass source. For this pressure, the highest total concentration of galactose and xylose as well as arabinose was observed: $169.8 \pm 0.6$ mg/g and $9.4 \pm 0.2$ mg/g for pine chips, $170.2 \pm 9.4$ mg/g and $22.0 \pm 0.2$ mg/g for beech chips and $119.2 \pm 1.2$ mg/g and $6.0 \pm 0.0$ for hemp stems, respectively (Figure 1A,C,E).

Compared to sulfuric acid, nitric acid used as a catalyst for degradation of lignocellulosic biomass exhibited lower efficiency in the cellulose decomposition process, which was manifested by a much lower concentration of glucose obtained from each type of plant biomass. The highest glucose concentration using this catalyst was observed only at the highest pressure used, i.e., 152 PSI ($52.6 \pm 0.2$, $33.2 \pm 2.8$ and $46.2 \pm 3.8$ mg/g for pine chips, beech chips and hemp stems, respectively; Figure 1B,D,F). In contrast to the process carried out with sulfuric acid, high levels of galactose and xylose, which are products of hemicellulose degradation, were reported with nitric acid over a wide pressure range: from 54 to 152 PSI. The lowest concentration of galactose and xylose using 1% *v/v* $HNO_3$ ($42.2 \pm 4.2$, $120.8 \pm 4.4$ and $31.6 \pm 0.8$ mg/g for pine chips, beech chips and hemp stems, respectively) was observed only for 152 PSI and the treatment time extended to 20 min (Figure 1B,D,F). The presented study clearly confirmed the usefulness of microwave pretreatment for the degradation of plant biomass, but the degree of degradation of individual carbohydrate fractions depended on selected process parameters, such as the pressure value and the type of acid catalyst used. In addition, it seems that obtaining some selectivity of biomass degradation (cellulose vs. hemicellulose) using microwaves could be possible. Hydrolysis using nitric acid in the first step of the process would promote hemicellulose

degradation. The resulting biomass could then be microwaved with sulfuric acid, which is more efficient in cellulose degradation. In this way, fractionation of sugars released during lignocellulosic biomass decomposition could be performed.

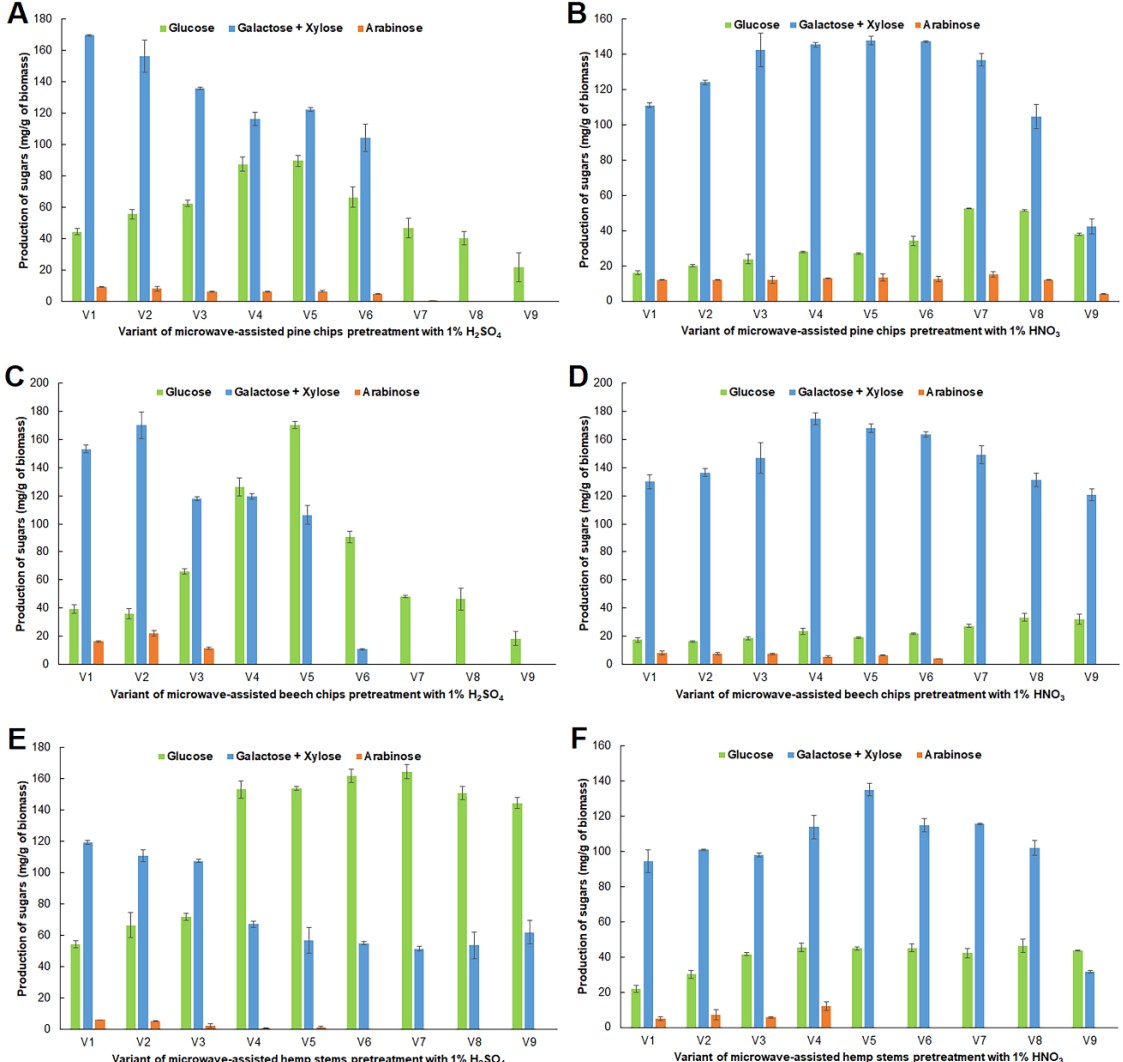

**Figure 1.** Concentrations of glucose, galactose/xylose and arabinose after microwave pretreatment in experimental variants ((**A**)—pine chips, 1% *v/v* $H_2SO_4$; (**B**)—pine chips, 1% *v/v* $HNO_3$; (**C**)—beech chips, 1% *v/v* $H_2SO_4$; (**D**)—beech chips, 1% *v/v* $HNO_3$; (**E**)—hemp stems, 1% *v/v* $H_2SO_4$; (**F**)—hemp stems, 1% *v/v* $HNO_3$).

To our knowledge, no comprehensive studies showing the impact of microwave treatment parameters on the degree of plant biomass decomposition have been presented so far. For this reason, we believe that the results of our work, which in detail show the impact of individual catalysts and process parameters on the degree of degradation of various types of biomass, fill a gap in the current knowledge. Ma et al. (2009) in their study on the possibility of rice straw degradation using microwave radiation showed, similarly to the present work, that the time of microwave treatment had little effect on the efficiency of both cellulose and hemicellulose degradation [21]. According to our own observations, pressure is a key parameter of microwave biomass pretreatment that provides a high degree of lignocellulose degradation. Determining the optimal pressure value, and hence the process temperature, is, next to the selection of the catalyst, the most important element of the degradation procedure setup. Similar observations were made by Orozco et al. (2007) and Teh et al. (2017), who showed that temperature is the most important parameter of microwave treatment guaranteeing

a high concentration of carbohydrates [21,22]. Orozco et al. (2007) reported that above 175 °C, concentrations of xylose and arabinose decreased [21]. In turn, Teh et al. (2017) showed that glucose and galactose levels decreased when the temperature exceeded 160 °C [22]. Our own studies as well as reports of other authors clearly indicate the need for precise optimization of microwave pretreatment parameters, such as pressure and catalyst type, in order to intensify the decomposition of plant biomass. Another way to isolate and selectively degrade hemicellulose is to use organic solvents such as methanol in combination with acid catalysis. Organosolv (3% $H_2SO_4$ and 40% *v/v* methanol) removed up to 75% hemicellulose from hemp shoots [23]. Pretreatment is also carried out using physical methods, e.g., radiation. Radiation strongly affects plant structures. Under the influence of this factor, solubility in hot water, ethanol and bases increases linearly. The content of structural components in plant fibers decreases with increasing radiation dose [24]. Research is also ongoing into the use of ionizing radiation in combination with the thermal or hydrothermal method. Duarte et al. (2012) reported a 20% increase in the sugarcane enzymatic hydrolysis efficiency using a 50 kGy dose in combination with the hydrothermal method and a 30% increase for the same dose in combination with acid catalysis [25]. To date, none of the available papers in the field has presented a comparison of the effects of microwave pretreatment using sulfuric or nitric acid on the type and amount of simple sugars obtained from various types of biomass. In this study, using microwave radiation, higher glucose concentrations were observed when sulfuric acid was used as compared to nitric acid. In contrast, the use of the latter caused more intense hemicellulose degradation, contributing to higher concentrations of galactose and xylose at elevated pressure (93 and 152 PSI). Dziekońska-Kubczak et al. (2018) using a conventional method of heating biomass did not find any differences in the concentration of glucose, xylose and arabinose for 2% $H_2SO_4$ and 2% $HNO_3$ [26]. These results suggest that the structural polysaccharide degradation may be affected by specific physical phenomena associated with the use of microwave radiation. In particular, it is believed that the oscillatory movement of dipoles, which causes hydrogen bond breakage in lignocellulose, plays an important role [8]. A higher glucose concentration observed in our studies when sulfuric acid was used rather than $HNO_3$ resulted from the greater effectiveness of sulfuric acid in cellulose degradation. However, the use of sulfuric acid in combination with elevated pressure (93 and 152 PSI) intensified the dehydration of sugars, i.e., glucose and xylose, which led to the formation of increased amounts of by-products (5-HMF and furfural). A higher concentration of galactose and xylose observed for nitric acid regardless of the type of biomass used is partly caused by the lack of conversion of xylose to furfural, because $HNO_3$ has lower ability to dehydrate carbohydrates [27]. Elevated concentrations of sugars (mainly xylose) released as a result of degradation of hemicellulose in a medium containing nitric acid may also result from the intensive breaking of glycosidic bonds in xylans composed mainly of β-1,4-xylopyranoses (β-1,4-D-xylopyranoses) by hydronium ions [28,29]. The effectiveness of microwave treatment combined with the use of dilute sulfuric acid was also confirmed by our own and other authors' previous research. It should be noted, however, that process parameters used for pine and beech chips or hemp stems, had a different effect on the amount of glucose produced when applied in the degradation of corn and wheat stillages. Regardless of the stillage used, there was no relationship between the maximum glucose concentration (ca. 60 mg/g and 100 mg/g DW of wheat and maize biomass, respectively), and the time and pressure of pretreatment at 300 W microwave generator power. The susceptibility of stillage biomass to decomposition, reported also for low pressure values and short exposure time, was probably due to previous barothermal treatment (from 73 to 88 PSI) during the preparation of the raw material with Henze steamer (release of starch grains from cellular structures) in the 1st generation ethanol production technology. On the other hand, as in the present study, the use of higher pressure of 152 PSI intensified sugar dehydration and promoted the formation of furfural, 5-HMF and levulinic acid [12,30]. High glucose levels during the degradation of sugarcane bagasse by microwave heating combined with the use of sulfuric acid were reported by Zhu et al. (2020). The use of 0.2 and 0.4 M sulfuric acid resulted in a higher concentration of glucose as a cellulose degradation product than when using alkaline catalysts; the latter promoted the degradation of hemicellulose rather than cellulose (higher levels of xylose and arabinose) [31].

## 2.2. Impact of Microwave Pretreatment Conditions on the Amount of Organic Acids, 5-HMF and Furfural Obtained from the Tested Plant Biomass

The effect of degradation of plant biomass using acid catalysts is a mixture of sugars that are products of cellulose and hemicellulose disintegration. However, inadequate selection of microwave pretreatment conditions, especially too high pressure, too long exposure time or excessive catalyst concentration, can lead to negative phenomena such as abundant formation and accumulation of biomass pretreatment by-products. As a result of microwave treatment of plant biomass, compounds recognized as inhibitors of metabolic processes are formed, including levulinic acid and 5-hydroxymethylfurfural (5-HMF), which are products of glucose dehydration, as well as acetic, formic acid and furfural (from xylose dehydration) [14,32]. The formation of acetic acid as a by-product of microwave pretreatment depends only on the type of raw material and not on the process parameters (Figure 2). The concentration of this acid remained at a constant level characteristic for a given biomass source: ca. 20 mg/g, 45–55 mg/g and 30–40 mg/g of pine chips, beech chips and hemp stems, respectively (Figure 2). Decreasing glucose, galactose and xylose levels as a result of microwave pretreatment of pine and beech chips using the highest pressure value (152 PSI) and 1% $v/v$ $H_2SO_4$ was associated with elevated concentrations of formic and levulinic acid (Figure 2A,C). Formic and levulinic acid concentrations were approximately 75 mg/g and 190 mg/g for pine chips, and ca. 68 mg/g and 140 mg/g for beech chips (Figure 2A,C). Nitric acid did not promote the dehydration of sugars such as glucose and xylose. For this reason, the concentration of formic acid was low (9–20 mg/g of biomass) and levulinic acid was not present at all in most solutions after microwave treatment regardless of the process conditions and the type of catalyst or raw material used (Figure 2B,D,F).

The occurrence of highly toxic sugar dehydration products, such as 5-HMF and furfural, depended primarily on the type of catalyst used and the pressure value during the microwave pretreatment. The use of sulfuric acid promoted the formation of 5-HMF and furfural, but the process conditions at which their highest concentrations were observed wee different for different types of plant biomass. Increased pressure (93 and 152 PSI) combined with 1% $v/v$ $H_2SO_4$ generated the highest furfural concentrations of: ca. 6 mg/g, 44 mg/g and 30 mg/g for pine chips, beech chips and hemp stems, respectively (Figure 3A,C,E). When nitric acid was used as a catalyst, only the highest pressure (152 PSI) during the exposure time extended to 20 min resulted in the formation of sugar dehydration products, i.e., 5-HMF and furfural (Figure 3B,D,F). The presented results confirmed the usefulness of microwave pretreatment in decomposition of plant biomass, however, special attention should be paid to the appropriate selection of process parameters. Too high pressure, and thus too high reaction temperature, can lead to the formation of toxic sugar dehydration products, whose elevated concentrations reduce the usefulness of the obtained hydrolysates for microbiological conversion processes.

The usefulness of microwave radiation was also demonstrated by other authors who even proposed this type of pretreatment for the production of compounds formed during the dehydration of sugars. Lacerda et al. (2015) used microwave radiation to produce 5-HMF and furfural, but the kinetics of the reaction depended primarily on the process temperature and the corresponding pressure. The amount of sugar dehydration products formed depended on the individual properties of the raw material, i.e., tropical palm trees of the genus *Copernicia* and *Acrocomia* [33]. In addition, Kłosowski et al. (2019) confirmed the possibility of using microwaves for plant biomass decomposition focused on the production of levulinic acid. Production efficiency was strongly dependent on the catalyst, pressure value and the type of lignocellulosic waste. In the process of converting sugars from softwood and hardwood biomass into levulinic acid using acid catalysts and microwaves, it is very important to choose the right acid to guarantee a high level of sugar dehydration. Sulfuric acid proved to be the most effective in converting glucose to levulinic acid. This was also confirmed by the results presented in this article, indicating a high degree of dehydration of sugars released from various raw materials as a result of the use of 1% $v/v$ $H_2SO_4$ under elevated pressure induced by microwaves. The pressure of 152 PSI during the acid decomposition of pine chips ensured a high concentration of

levulinic acid (ca.190 mg/G DW) regardless of the exposure time. This is in line with the results of the study on the targeted conversion of glucose to levulinic acid at a pressure of 225 PSI [10].

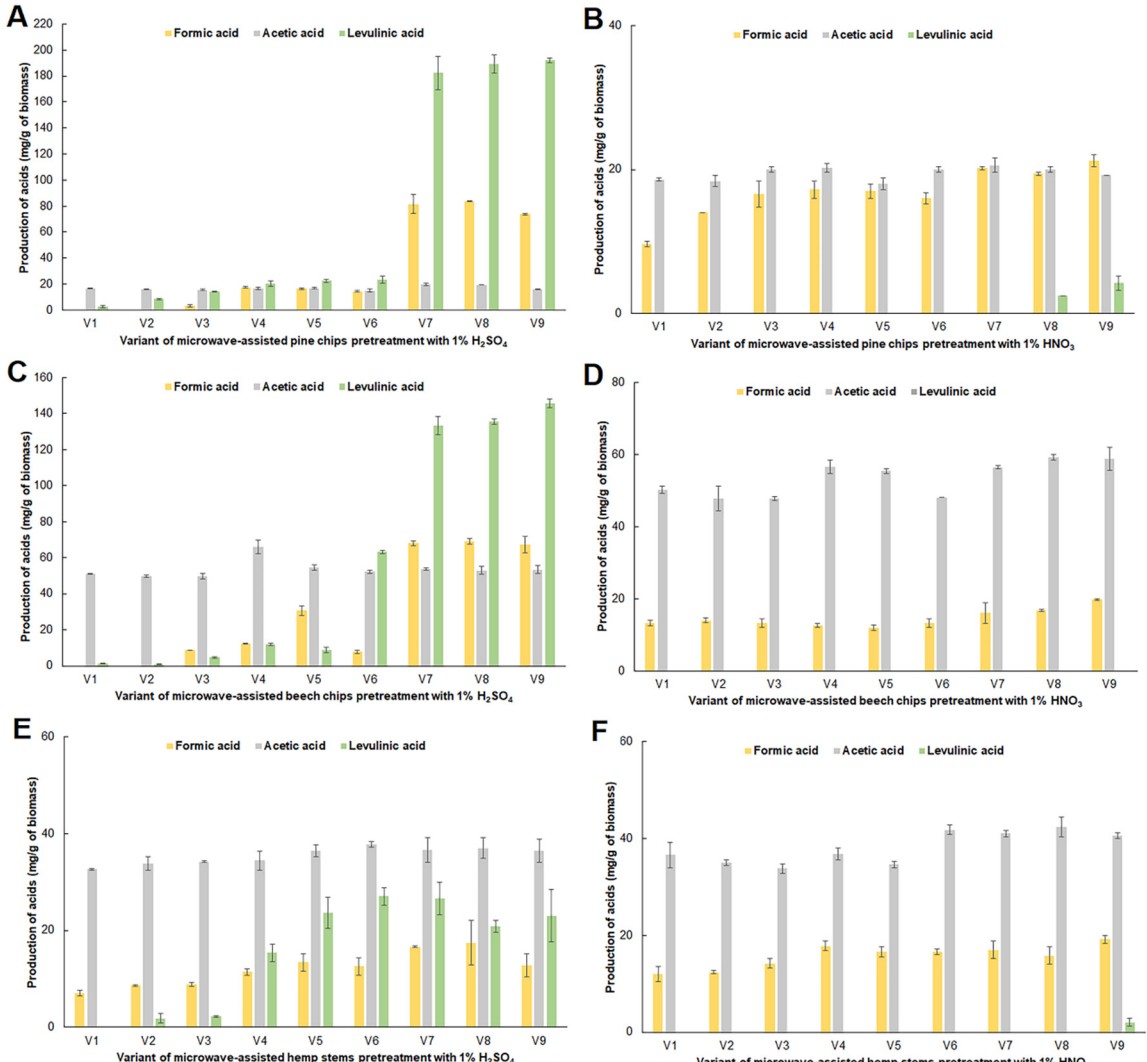

**Figure 2.** Concentrations of organic acids after microwave pretreatment in experimental variants ((**A**)—pine chips, 1% *v/v* H$_2$SO$_4$; (**B**)—pine chips, 1% *v/v* HNO$_3$; (**C**)—beech chips, 1% *v/v* H$_2$SO$_4$; (**D**)—beech chips, 1% *v/v* HNO$_3$; (**E**)—hemp stems, 1% *v/v* H$_2$SO$_4$; (**F**)—hemp stems, 1% *v/v* HNO$_3$).

The effectiveness of acid dehydration of sugars using microwaves was also confirmed in studies using HCl for the production of 5-HMF and levulinic acid from sugarcane bagasse. The effective production of levulinic acid was favored by high temperature (170–190 °C), however the highest concentration of 5-HMF was reported for 130 °C, i.e., the lowest temperature from those analyzed using HCl as a catalyst [34]. In addition, in the present study on the use of sulfuric acid in pine chip degradation, the highest concentration of 5-HMF was obtained at moderate pressures of 54 and 93 PSI. Stronger impact of sulfuric acid on dehydration of carbohydrates and their conversion to toxic by-products have also been confirmed in studies on cellulosic ethanol production from Jerusalem artichoke stalks. The use of sulfuric acid at a concentration as low as 2% *v/v* resulted in higher concentrations of 5-HMF and furfural compared to the same concentration of nitric acid [26].

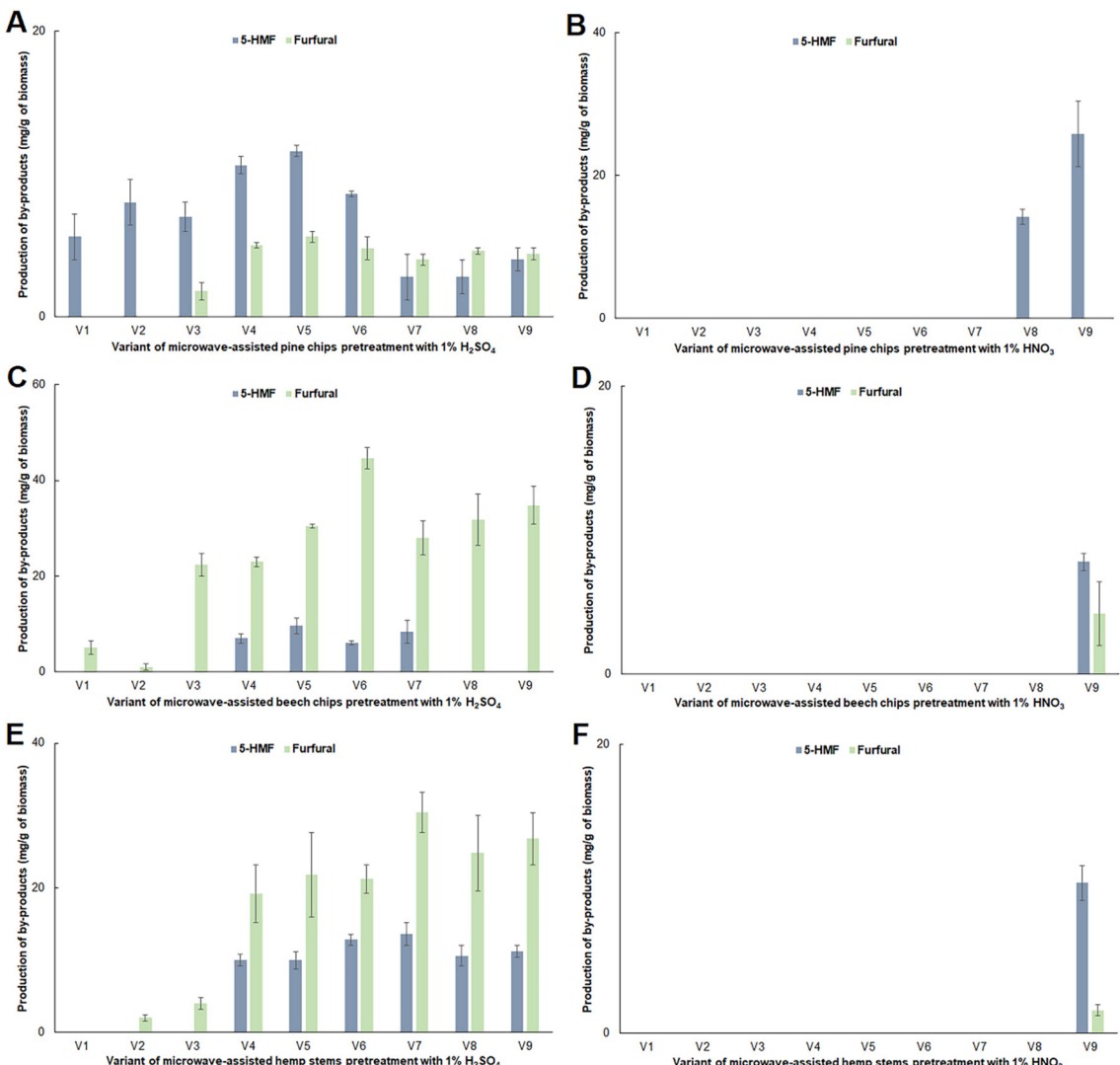

**Figure 3.** Concentrations of 5-hydroxymethylfurfural (5-HMF) and furfural after microwave pretreatment in experimental variants ((**A**)—pine chips, 1% *v/v* $H_2SO_4$; (**B**)—pine chips, 1% *v/v* $HNO_3$; (**C**)—beech chips, 1% *v/v* $H_2SO_4$; (**D**)—beech chips, 1% *v/v* $HNO_3$; (**E**)—hemp stems, 1% *v/v* $H_2SO_4$; (**F**)—hemp stems, 1% *v/v* $HNO_3$).

## 3. Materials and Methods

### 3.1. Characteristics of Raw Materials

As sources of lignocellulose, we used pine chips (softwood), beech chips (hardwood) and hemp stems (a non-wood raw material). The content of cellulose, hemicellulose and lignin as well as dry matter content in raw materials is presented in Table 1. Waste soft- and hardwood was supplied by Radex sawmill (Wudzyń, Poland). Pine and beech chips were shredded using a beater mill and sieved through a 1.0 mm sieve. Hemp stems were provided by the MyOrganico farm (Poland), dried and ground using a beater mill.

**Table 1.** Composition of plant biomass used in this study.

| Components | Pine Chips | Beech Chips | Hemp Stems |
|---|---|---|---|
| Dry weight [%] | 93.07a ± 0.12 | 93.63a ± 0.13 | 91.19b ± 0.07 |
| Content of cellulose [% dry weight] | 49.93a ± 0.87 | 56.24b ± 1.14 | 67.16c ± 1.91 |
| Content of hemicelluloses [% dry weight] | 14.33a ± 2.49 | 22.67b ± 1.45 | 11.83a ± 1.93 |
| Content of lignin [% dry weight] | 26.68a ± 0.27 | 14.67b ± 0.65 | 11.33c ± 0.29 |

The mean values given in lines with different letter index are significantly different ($\alpha < 0.05$).

### 3.2. Chemical Reagents

All reagents used in the study were of analytical purity and were provided by Merc® (Darmstadt, Germany). Chromatographic analyzes were performed using HPLC grade solvents supplied by Merck®. Calibration standards for chromatographic analyzes were HPLC grade and were provided by Sigma-Aldrich® (St. Louis, MO, USA).

### 3.3. Selection of Conditions for Microwave Degradation of Plant Biomass

The microwave pretreatment was carried out using the Microwave Digestion System Mars 5 (CEM Corporation) which enabled the process to be carried out under controlled pressure and temperature conditions. The pretreatment procedure was started by placing 1 g DW (=1.05 g of biomass) in a Teflon container HP-500 plus, then 20 mL 1% *v/v* $H_2SO_4$ or 1% *v/v* $HNO_3$ were added. The microwave pretreatment process was conducted under various pressure conditions (54, 93, 152 PSI) and varying pretreatment time (10, 15, 20 min). The assumption was that the pressure used was 54, 93 or 152 PSI, which corresponded to approximately 140, 160 and 180 °C, and was kept constant throughout the entire biomass pretreatment process. Table 2 presents the characteristics of all combinations of process parameters that made up 9 experimental variants for each type of catalyst. The samples were heated using constant power of microwave generator (300 W). After the pretreatment, the solutions were cooled down to approximately 20 °C, quantitatively transferred to 50 mL laboratory beakers and neutralized to pH 5.50 ± 0.05 with 30% NaOH. The volume was made up with deionized water to 40 mL. The solutions prepared in this way were sampled to determine (by HPLC) the concentration of carbohydrates, organic acids and 5-HMF and furfural.

**Table 2.** Experimental variants used in the selection of optimal conditions of microwave pretreatment.

| Biomass Pretreatment Variant | Power of the Microwave Generator [W] | Pressure [PSI] | Pretreatment Time [min] |
|---|---|---|---|
| V1 | 300 | 54 | 10 |
| V2 | 300 | 54 | 15 |
| V3 | 300 | 54 | 20 |
| V4 | 300 | 93 | 10 |
| V5 | 300 | 93 | 15 |
| V6 | 300 | 93 | 20 |
| V7 | 300 | 152 | 10 |
| V8 | 300 | 152 | 15 |
| V9 | 300 | 152 | 20 |

### *3.4. Analytical Methods*

### 3.4.1. Biomass Characterization

Cellulose, hemicellulose and lignin content were determined using two-stage acid hydrolysis according to the NREL protocol [35]. The biomass components were determined after acid hydrolysis, so that HPLC analysis could be performed. Structural polysaccharides were hydrolyzed to simple sugars and their concentration was determined using chromatographic techniques. Acid soluble lignin was determined after extraction using a UV–Vis spectrophotometer (Pharo 300 by Merck) at 240 nm. The dry weight (DW) was determined using a weighting dryer (Radwag WPS-30S). Parameters of the drying process: temperature 130 °C, sampling time 20 s.

### 3.4.2. Determination of Carbohydrates and Organic Acid after Microwave-Assisted Pretreatment

Prior to the HPLC analysis, samples were diluted 5-fold in the mobile phase (5 mM $H_2SO_4$) and filtered through a membrane filter (pore size 0.45 μm). The analysis was carried out using an Agilent Technologies® chromatograph (model 1260) equipped with a refractometric detector. Chromatographic separation was performed using a Hi-Plex H column (Agilent Technologies®) equipped with a dedicated guard column, with a mobile phase flow rate of 0.6 mL/min at 60 °C. Quantitative analysis was performed based on ESTD calibration. The Hi-Plex H column did not allow for effective separation of xylose and galactose, therefore the concentration of these sugars was given as the total concentration of both compounds [36].

### 3.4.3. Determination of 5-HMF and Furfural

Samples for analysis were diluted and filtered through a 0.22 μm membrane filter. The analysis was performed using an Agilent Technologies® chromatograph (model 1260) equipped with a diode detector (DAD). The separation was performed with a ZORBAX Eclipse Plus C18 (4.6 × 100 mm, 3.5 μm) column (Agilent Technologies®) using 0.3% acetic acid (70%) and methanol (30%) as a mobile phase with a flow rate of 0.5 mL/min at 30 °C. Furfural and 5-HMF were detected at 280 nm [37]. Quantitative analysis was performed using the ESTD method.

### *3.5. Statistics*

All laboratory analyses were performed in triplicate. Statistical analysis was carried out using the Statistica software ver. 12 (analysis of variance, determination of SD). ANOVA test and HSD (honest significant difference) Tukey's test were applied at the significance level of $\alpha < 0.05$.

## 4. Conclusions

Microwave radiation is an effective way to support the decomposition of various types of plant biomass, providing high concentrations of simple sugars such as glucose, galactose and xylose. However, the use of this pretreatment method must be associated with precise optimization of process parameters (pressure and exposure time), and selection of an acid catalyst. Diluted sulfuric and nitric acid can be used as a catalyst in the process of microwave decomposition of softwood, hardwood and non-wood biomass, giving hydrolysates with a high concentration of simple sugars for use in biosynthesis processes. However, the use of microwave radiation to degrade lignocellulose requires detailed research on the optimization of parameters specific for each raw material, guaranteeing not only hydrolysates with a high concentration of carbohydrates, but also a low concentration of fermentation inhibitors. It should be emphasized, however, that the pressure at which the reaction is conducted is an extremely important process parameter that requires a special control. A precise selection of this parameter value intensifies the release of simple sugars, which can be used in subsequent microbial conversion processes, e.g., in the production of renewable energy carriers. The results indicated the high potential of microwaves used in the lignocellulose decomposition process. For this reason, studies

on this method of heating biomass in combination with the use of other factors that may increase the degree of degradation of structural polysaccharides (e.g., hydrotropes) seem worth continuing.

**Author Contributions:** G.K. and D.M. conceived and designed the experiments; D.M. and N.L. performed the experiments; G.K. and D.M. analyzed the data; D.M. and N.L. contributed reagents/materials/analysis tools; G.K. and D.M. wrote the paper. All authors have read and agreed to the published version of the manuscript.

**Funding:** This research was funded by the Polish Minister of Science and Higher Education, under the program "Regional Initiative of Excellence" in 2019—2022 grant number 008/RID/2018/19 and the APC was funded by the program "Regional Initiative of Excellence" in 2019—2022 (Grant No. 008/RID/2018/19).

**Conflicts of Interest:** The authors declare no conflicts of interest.

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
