# Peer review of "Microwave-Assisted Degradation of Biomass with the Use of Acid Catalysis"

_catalysts, doi:10.3390/catal10060641_

Round 1

Reviewer 1 Report

Manuscript ID- catalysts-802928 Title-Microwave-assisted degradation of biomass with the use of acid catalysis

Manuscript presents very good research related to microwave driven degradation of biomass and going to be interesting for the readers. It required minor correction before consideration to publication.

  • Authors need to include some interesting data in the abstract part of the manuscript.
  • Introduction part looks too short authors need to improve it with recent references.
  • Authors need to include some important references related to acid driven degradation of biomass and their applications.
  • Authors need to compare their results with previously reported similar research.
  • Authors need to add future prospective of the presented research in the conclusion part of the manuscript.

Author Response

Detailed response to the Reviewers' comments (catalysts-802928)

We would like to thank the Editor and the Reviewers for their comments and suggestions. We have revised the manuscript point by point according to the Reviewers’ comments. All the suggested changes are marked in yellow in the revised text and described below.

We hope that the quality and readability of our manuscript have been improved.

Response to Reviewer No. 1 comments:

Manuscript presents very good research related to microwave driven degradation of biomass and going to be interesting for the readers. It required minor correction before consideration to publication.

Authors need to include some interesting data in the abstract part of the manuscript.

According to the Reviewer’s suggestion abstract has been supplemented with additional data (lines 18-21).

Introduction part looks too short authors need to improve it with recent references. Authors need to include some important references related to acid driven degradation of biomass and their applications.

Additional information about dilute acid pretreatment has been added in Introduction Section (lines 62-74).

Authors need to compare their results with previously reported similar research.

According to the Reviewer’s suggestion discussion section has been revised in lines 152-161, 168-188, 239-243.

Authors need to add future prospective of the presented research in the conclusion part of the manuscript.

According to the Reviewer’s suggestion Conclusion Section has been revised (lines 324-327).

Reviewer 2 Report

The manuscript reports  interesting results obtained with different type of biomass. However discussion is almost totaly absent. I would appreciate comparison with other methods such as organosolv (ChemSusChem 2014, 7, 1991 – 1999 for example in the case of hemp stems)

An evaluation of the energy demand of the proposed MW promoted process is also totally lacking.

Author Response

Detailed response to the Reviewers' comments (catalysts-802928)

We would like to thank the Editor and the Reviewers for their comments and suggestions. We have revised the manuscript point by point according to the Reviewers’ comments. All the suggested changes are marked in yellow in the revised text and described below.

We hope that the quality and readability of our manuscript have been improved.

Response to Reviewer No. 2 comments:

The manuscript reports  interesting results obtained with different type of biomass. However discussion is almost totaly absent. I would appreciate comparison with other methods such as organosolv (ChemSusChem 2014, 7, 1991 – 1999 for example in the case of hemp stems).

As suggested by the reviewer, the discussion has been supplemented with a comparison of microwave treatment with organosolv technique, which is another pretreatment method (lines 152-161).

An evaluation of the energy demand of the proposed MW promoted process is also totally lacking.

As the Reviewer rightly noted, it is important to provide the energy demand for the process of biomass degradation using microwaves, however, learning the exact value of this parameter will be possible only when using a fully metered reactor prototype. The construction of such a reactor is planned in further stages of research on catalysis using microwave radiation.

Reviewer 3 Report

General Comments:

The manuscript addresses an experimental and systematic study about the degradation of different types of biomass using nitric and sulfuric acid under microwave irradiation. The variables studied were time, pressure (for each acid and type of biomass). The work contains very interesting results although the manuscript fails in the experimental planning and the discussion is weak and purely descriptive.

According to other published papers, temperature is a key factor in the depolymerisation of lignocellulosic biomass as mentioned in the manuscript. However, authors do not report the temperature reached in each treatment. Was temperature measured in each experiment? Does temperature depend on pressure and/or time? Was temperature affected by type of acid or type of biomass? Was temperature constant along the reaction time? All these questions are not answered in the manuscript and they are critical to understand and discuss the results properly.

There is a huge amount of papers dealing with microwave assisted acid degradation of biomass. Therefore, authors should clearly state the novelty of the manuscript and the most remarkable contribution of the work to the existing literature in the field.

As mentioned above, the discussion is poor because it is merely a description of the results, pointing the best experiment and the trend for each figure, without trying to explain the reason behind those results and without a deep comparison with other published papers. Why sulfuric acid produce different product distribution than that produced with nitric acid? These type of questions should be answer during the discussion and not only state the facts.

The experimental plot is nice but authors could get more out of the results if they analyse the results in the base of a powerful statistical tool such as the experimental design of experiments. Actually, authors performed a full two-factor three-level factorial design (by triplicate) as it can be seen in Table 2. Therefore, they obtain mathematical models for each response selected (the production of each compound for each type of acid and type of biomass) and analyse the results much deeper and comprehensively with the aid of the generated response surfaces. This would enrich the discussion section.

Therefore, the manuscript is not adequate to be published in Catalysts journal in the present form.

Minor Comments:

(1) Figures 1, 2 and 3: The y-axis in all figures do not represent any concentration but the production of glucose, formic acid, 5-HMF, etc. per gram of dry biomass for each treatment.

(2) Although correctly cited, some details about the basis of the analytical protocols for biomass characterisation shold be added in section 3.4.1.

Author Response

Detailed response to the Reviewers' comments (catalysts-802928)

We would like to thank the Editor and the Reviewers for their comments and suggestions. We have revised the manuscript point by point according to the Reviewers’ comments. All the suggested changes are marked in yellow in the revised text and described below.

We hope that the quality and readability of our manuscript have been improved.

Response to Reviewer No. 3 comments:

General Comments:

The manuscript addresses an experimental and systematic study about the degradation of different types of biomass using nitric and sulfuric acid under microwave irradiation. The variables studied were time, pressure (for each acid and type of biomass). The work contains very interesting results although the manuscript fails in the experimental planning and the discussion is weak and purely descriptive.

According to other published papers, temperature is a key factor in the depolymerisation of lignocellulosic biomass as mentioned in the manuscript. However, authors do not report the temperature reached in each treatment. Was temperature measured in each experiment? Does temperature depend on pressure and/or time? Was temperature affected by type of acid or type of biomass? Was temperature constant along the reaction time? All these questions are not answered in the manuscript and they are critical to understand and discuss the results properly.

Indeed, elevated temperature, and hence also elevated pressure, is a key parameter affecting the efficiency of lignocellulosic biomass degradation. Due to the dependence of vapor pressure on temperature, the process of lignocellulose degradation using microwaves can be controlled by either temperature or pressure. Depending on the type of acid used, the process temperature at the same pressure will vary, and vice versa. When conducting the experiments, the authors focused on the strict pressure control using the Mars 5 device by CEM, because pressure control is easier in industrial conditions. The assumption was that the pressure used was 54, 93 or 152 PSI, which corresponded to approximately 140, 160 and 180ºC, and was kept constant throughout the entire biomass pretreatment process. An appropriate explanation has been placed in the text in lines 272-274.

There is a huge amount of papers dealing with microwave assisted acid degradation of biomass. Therefore, authors should clearly state the novelty of the manuscript and the most remarkable contribution of the work to the existing literature in the field.

The Reviewer's claim that there is a huge amount of papers describing the impact of microwave treatment on the degradation of plant biomass is probably based on some misunderstanding. The authors thoroughly analyzed the available literature in the field. There are no publications describing comprehensively the influence of the parameters of microwave pretreatment of various types of biomass on the amount of simple sugars released and the resulting by-products. If any of the available papers in this field have escaped our attention, we will sincerely appreciate reading suggestions. The authors, as suggested by the reviewer, have revised the description of the novelty of the presented research (lines 88-94).

As mentioned above, the discussion is poor because it is merely a description of the results, pointing the best experiment and the trend for each figure, without trying to explain the reason behind those results and without a deep comparison with other published papers. Why sulfuric acid produce different product distribution than that produced with nitric acid? These type of questions should be answer during the discussion and not only state the facts.

The discussion section has been revised as suggested by the Reviewer in lines 152-161, 168-188, 239-243.

The experimental plot is nice but authors could get more out of the results if they analyse the results in the base of a powerful statistical tool such as the experimental design of experiments. Actually, authors performed a full two-factor three-level factorial design (by triplicate) as it can be seen in Table 2. Therefore, they obtain mathematical models for each response selected (the production of each compound for each type of acid and type of biomass) and analyse the results much deeper and comprehensively with the aid of the generated response surfaces. This would enrich the discussion section.

As the Reviewer noted, in such an experiment it is possible to perform statistical analysis containing mathematical models illustrating the results obtained. However, to enable the creation of a mathematical model, a linear selection of parameter values is necessary, which is not the case in the presented work. The authors agree with the Reviewer's suggestion and in the next stages of the research plan to use advanced statistical tools providing a more complete description of the results. In addition, the authors intend to purchase the specialized Statistical Software Design-Expert ver.11 to design experiments using advanced statistical tools such as two-level factor planning or general factor analysis.

Therefore, the manuscript is not adequate to be published in Catalysts journal in the present form.

Minor Comments:

(1) Figures 1, 2 and 3: The y-axis in all figures do not represent any concentration but the production of glucose, formic acid, 5-HMF, etc. per gram of dry biomass for each treatment.

As suggested by the Reviewer, all Figures have been corrected.

(2) Although correctly cited, some details about the basis of the analytical protocols for biomass characterisation should be added in section 3.4.1.

As suggested by the reviewer, the Biomass characterization section has been revised (lines 286-290).

Round 2

Reviewer 2 Report

The manuscript has very much improved.

It can be accepted without further modification

Author Response

The authors agree with the reviewer.

Reviewer 3 Report

Authors have answered the comment about the effect of temperature and about the experimental design satisfactorily.

Concerning the comment in which I suggest the authors to describe the novelty of the manuscript, I must apologise for a typing mistake because I should have said, "There is a huge amount of papers dealing with microwave assisted OR acid degradation of biomass". Authors correctly pointed out that there are not many papers dealing with simultaneous microwave and acid treatment of biomass. I meant that there are many papers about acid treatment and many about microwave treatment (for different type of biomass, catalysts, solvents, etc.). These studies can be used to compare and stress the advantages of simultaneously use MW and acids. However, still some recent references about simultaneous treatments can be helpful to authors. See, for example:

(1) Sangib, EB et al. (2020). Optimization of cellulose hydrolysis in the presence of biomass-derived sulfonated catalyst in microwave reactor using response surface methodology. BIOMASS CONVERSION AND BIOREFINERY (DOI: 10.1007/s13399-020-00720-2).

(2) Shao, Y et al. (2020). Acidic seawater improved 5-hydroxymethylfurfural yield from sugarcane bagasse under microwave hydrothermal liquefaction. ENVIRONMENTAL RESEARCH, 184, 109340 (DOI: 10.1016/j.envres.2020.109340).

(3) Zhu, Z et al. (2020). Comparative evaluation of microwave-assisted acid, alkaline, and inorganic salt pretreatments of sugarcane bagasse for sugar recovery. BIOMASS CONVERSION AND BIOREFINERY (DOI: 10.1007/s13399-020-00680-7).

(4) Bhardwaj, N et al. (2020) Microwave-assisted pretreatment using alkali metal salt in combination with orthophosphoric acid for generation of enhanced sugar and bioethanol. BIOMASS CONVERSION AND BIOREFINERY (10.1007/s13399-020-00640-1).

(5) Sorn, V et al. (2019). Effect of microwave-assisted ionic liquid/acidic ionic liquid pretreatment on the morphology, structure, and enhanced delignification of rice straw. BIORESOURCE TECHNOLOGY, 293, 121929 (DOI: 10.1016/j.biortech.2019.121929).

(6) Thangavelu, S et al. (2019). Microwave assisted acid hydrolysis for bioethanol fuel production from sago pith waste. WASTE MANAGEMENT, 86, 80-86 (DOI: 10.1016/j.wasman.2019.01.035).

(7) Carrion-Prieto, P et al. (2018). Furfural, 5-HMF, acid-soluble lignin and sugar contents in C. ladanifer and E. arborea lignocellulosic biomass hydrolysates obtained from microwave-assisted treatments in different solvents. BIOMASS & BIOENERGY, 119, 135-143 (DOI: 10.1016/j.biombioe.2018.09.023).

(8) Mikulski, D et al. (2020). Microwave-assisted dilute acid pretreatment in bioethanol production from wheat and rye stillages. BIOMASS & BIOENERGY, 136, 105528 (DOI: 10.1016/j.biombioe.2020.105528).

Looking more carefully into the references cited in the manuscript (especially references 10 and 12) and reference 8 (above), one can see that most of the conclusions from this manuscript are not new and are similar to those obtained in the mentioned previous studies (Cited references 10, 12 and suggested reference 8 share two of the authors with this manuscript).

In cited reference 10, authors studied the hydrolysis of lignocellulosic biomass (wheath and rye stillages) using the same experimental planning, i.e., Pressure (54, 93 and 152 psi), time (10, 15, 20 min), power (300 W) and sulfuric acid and biomass concentration. The work also measured the production of glucose, galactose, arabinose acetic acid, levullinic acid, HMF and furfural. This paper share important results and conclusions with the present manuscript that have not been discussed. This paper was only cited in the introduction and, in a general manner, in the discussion.

A similar situation happens again with respect cited reference 12. In this case, other lignocellulosic biomass (maize stillage) was used but similar conditions of the microwave-acid treatment were applied. Again, this paper was only cited in the introduction although the work must be mentioned during the discussion of the results.

There is a third article (suggested reference 8 above) where authors studied the same biomass (pine wood chips) to produce levullinic acid in the same experimental system under common conditions with this manuscript. In fact, Table 5 of that paper contains the same experimental values for the biomass characterisation that those reported in the present manuscript in Table 1 but without citation. In that work, pine chips were subjected to MW-acid treatment with 5 and 10% biomass, 54 psi, 1 and 2% of sulfuric and nitric acids. The only difference was the power of the MW source (1200 W). Once again, the citation of that closely related paper is not cited correctly in the discussion of the manuscript.

All the above papers and this new manuscript reach similar partial conclusions because they share similar experimental set-up or raw materials or degradation products. Therefore, the novelty of the manuscript remains unclear and authors should define clearly the differences with respect those three papers.

In addition, authors claim (in the conclusions section) that "Microwave radiation is an effective way to support the decomposition of various types of plant biomass, ...", "The use of sulfuric acid together with microwaves increases the decomposition of various types of plant biomass." but these conclusions can be obtained from other previous papers.

Hence, I feel that authors have not clearly stated the advances and novelties achieved in the manuscript, which is key to be acceptable for publication in any scholar journal.

Author Response

Detailed response to the Reviewer’s comments (catalysts-802928)

We would like to thank the Editor and the Reviewer for their comments and suggestions. We have revised the manuscript point by point according to the Reviewer’s comments. All the suggested changes are marked in yellow in the revised text and described below.

We hope that the quality and readability of our manuscript have been improved.

Response to Reviewer No. 3 comments:

Authors have answered the comment about the effect of temperature and about the experimental design satisfactorily.

Concerning the comment in which I suggest the authors to describe the novelty of the manuscript, I must apologise for a typing mistake because I should have said, "There is a huge amount of papers dealing with microwave assisted OR acid degradation of biomass". Authors correctly pointed out that there are not many papers dealing with simultaneous microwave and acid treatment of biomass. I meant that there are many papers about acid treatment and many about microwave treatment (for different type of biomass, catalysts, solvents, etc.). These studies can be used to compare and stress the advantages of simultaneously use MW and acids. However, still some recent references about simultaneous treatments can be helpful to authors.

Looking more carefully into the references cited in the manuscript (especially references 10 and 12) and reference 8 (above), one can see that most of the conclusions from this manuscript are not new and are similar to those obtained in the mentioned previous studies (Cited references 10, 12 and suggested reference 8 share two of the authors with this manuscript).

In cited reference 10, authors studied the hydrolysis of lignocellulosic biomass (wheath and rye stillages) using the same experimental planning, i.e., Pressure (54, 93 and 152 psi), time (10, 15, 20 min), power (300 W) and sulfuric acid and biomass concentration. The work also measured the production of glucose, galactose, arabinose acetic acid, levullinic acid, HMF and furfural. This paper share important results and conclusions with the present manuscript that have not been discussed. This paper was only cited in the introduction and, in a general manner, in the discussion.

A similar situation happens again with respect cited reference 12. In this case, other lignocellulosic biomass (maize stillage) was used but similar conditions of the microwave-acid treatment were applied. Again, this paper was only cited in the introduction although the work must be mentioned during the discussion of the results.

As suggested by the Reviewer, we have supplemented the discussion with papers presenting the impact of microwave radiation on the degradation of waste stillage biomass (lines 193-209).

There is a third article (suggested reference 8 above) where authors studied the same biomass (pine wood chips) to produce levullinic acid in the same experimental system under common conditions with this manuscript. In fact, Table 5 of that paper contains the same experimental values for the biomass characterisation that those reported in the present manuscript in Table 1 but without citation. In that work, pine chips were subjected to MW-acid treatment with 5 and 10% biomass, 54 psi, 1 and 2% of sulfuric and nitric acids. The only difference was the power of the MW source (1200 W). Once again, the citation of that closely related paper is not cited correctly in the discussion of the manuscript.

The discussion covering the formation of sugar dehydration products has been supplemented as suggested by the Reviewer (lines 259-268).

All the above papers and this new manuscript reach similar partial conclusions because they share similar experimental set-up or raw materials or degradation products. Therefore, the novelty of the manuscript remains unclear and authors should define clearly the differences with respect those three papers.

As suggested by the Reviewer, the novelty of the presented study has been emphasized in relation to those already published (lines 94-105). It should be noted that high process efficiency requires careful optimization of process parameters, separately for each type of raw material. These and previous results achieved for various lignocellulosic materials indicate that it is practically impossible to propose universal conditions for the pretreatment process that could be successfully used regardless of the type of plant biomass. The material-dependent biomass preprocessing is not just the choice of pretreatment conditions that ensure high hydrolysis efficiency. This approach also reduces or eliminates the adverse formation of excessive amounts of inhibitors.

In addition, authors claim (in the conclusions section) that "Microwave radiation is an effective way to support the decomposition of various types of plant biomass, ...", "The use of sulfuric acid together with microwaves increases the decomposition of various types of plant biomass." but these conclusions can be obtained from other previous papers.

The Conclusion section has been revised as suggested by the Reviewer in lines 359-365.

Hence, I feel that authors have not clearly stated the advances and novelties achieved in the manuscript, which is key to be acceptable for publication in any scholar journal.